

# Evaluating multi-locus phylogenies for species boundaries determination in the genus *Diaporthe*

Liliana Santos[1], Artur Alves[1] and Rui Alves[2]

[1] Departamento de Biologia, CESAM, Universidade de Aveiro, Aveiro, Portugal
[2] Departament de Ciències Mèdiques Bàsiques, Universitat de Lleida and IRBLleida, Lleida, Spain

## ABSTRACT

**Background**. Species identification is essential for controlling disease, understanding epidemiology, and to guide the implementation of phytosanitary measures against fungi from the genus *Diaporthe*. Accurate *Diaporthe* species separation requires using multi-loci phylogenies. However, defining the optimal set of loci that can be used for species identification is still an open problem.

**Methods**. Here we addressed that problem by identifying five loci that have been sequenced in 142 *Diaporthe* isolates representing 96 species: *TEF1*, *TUB*, *CAL*, *HIS* and ITS. We then used every possible combination of those loci to build, analyse, and compare phylogenetic trees.

**Results**. As expected, species separation is better when all five loci are simultaneously used to build the phylogeny of the isolates. However, removing the ITS locus has little effect on reconstructed phylogenies, identifying the *TEF1-TUB-CAL-HIS* 4-loci tree as almost equivalent to the 5-loci tree. We further identify the best 3-loci, 2-loci, and 1-locus trees that should be used for species separation in the genus.

**Discussion**. Our results question the current use of the ITS locus for DNA barcoding in the genus *Diaporthe* and suggest that *TEF1* might be a better choice if one locus barcoding needs to be done.

## INTRODUCTION

Species in the ascomycete genus *Diaporthe* have been identified all over the world. Typically, *Diaporthe* species are saprobes, endophytes, or plant pathogens (*Webber & Gibbs, 1984*; *Boddy & Griffith, 1989*; *Udayanga et al., 2011*). Some plant pathogenic *Diaporthe* species are associated with cankers, diebacks, rots, spots and wilts on a wide range of plants, some of which are of economic importance as is the case of citrus, cucurbits, soybeans, eggplant, berries and grapevines (*Backman, Weaver & Morgan-Jones, 1985*; *Merrin, Nair & Tarran, 1995*; *Farr, Castlebury & Rossman, 2002*; *Farr et al., 2002*; *Shishido et al., 2006*). Less frequently, *Diaporthe* species can also cause lupinosis and other health problems in humans and other mammals (*Van Warmelo & Marasas, 1972*; *Sutton et al., 1999*; *Battilani et al., 2011*; *Garcia-Reyne et al., 2011*).

Corresponding authors
Artur Alves, artur.alves@ua.pt
Rui Alves, ralves@cmb.udl.cat

Distinction between *Diaporthe* species has historically been based on an approach that combined morphological information, cultural characteristics, and host affiliation (*Udayanga et al., 2011*). This approach made it difficult to reliably discriminate between the various members of the genus, because many of these fungi are asexual with low host specificity (*Rehner & Uecker, 1994*; *Murali, Suryanarayanan & Geeta, 2006*). As a consequence, an unnecessary increase in the number of proposed *Diaporthe* species occurred. This number currently stands at 977 and 1,099 for *Diaporthe* and 980 and 1,047 for *Phomopsis* in Index Fungorum (http://www.indexfungorum.org/) and Mycobank (http://www.mycobank.org/), respectively (both accessed 14 November 2016). The extinction of the dual nomenclature system for fungi raised the question about which generic name to use, *Diaporthe* or that of its asexual morph *Phomopsis*. Given that both names are well known among plant pathologists, and have been equally used, *Rossman et al. (2015)* proposed that the genus name *Diaporthe* should be retained over *Phomopsis* because it was introduced first and therefore has priority.

The problem of incorrect species attribution has practical consequences for the study of this genus, because accurate species identification is essential for understanding the epidemiology, for controlling plant diseases, and to guide the implementation of international phytosanitary measures (*Santos & Phillips , 2009*; *Udayanga et al., 2011*). Therefore, there was an urgent need to reformulate species identification in the genus *Diaporthe* (*Santos & Phillips , 2009*).

Advances in the areas of gene sequencing and molecular evolution over the last 50 years have led to the notion that ribosomal genes can be used to distinguish between species and study their molecular evolution (*Woese & Fox, 1977*). The choice of these genes comes from the fact that their function is conserved over all living organisms, which has been assumed to imply that their evolutionary rate should be roughly constant over the tree of life.

The molecular evolution studies mentioned have been used to develop general fungal classifications (*Shenoy, Jeewon & Hyde, 2007*) and have also been used for species reclassification in the genus *Diaporthe* (*Santos & Phillips , 2009*; *Santos et al., 2011*; *Thompson et al., 2011*; *Baumgartner et al., 2013*; *Gomes et al., 2013*; *Huang et al., 2013*; *Tan et al., 2013*; *Gao et al., 2014*; *Udayanga et al., 2014a*; *Udayanga et al., 2014b*). In fact, recently the ITS region of the ribosomal genes has been accepted as the official fungal barcode (*Schoch et al., 2012*), and its sequence is frequently used for molecular phylogeny analysis of *Diaporthe* species.

However, assuming that ribosomal gene sequences evolve at a uniform rate, independent of species is sometimes incorrect (*Anderson & Stasovski, 1992*; *O'Donnell, 1992*; *Carbone & Kohn, 1993*). In addition, due to the strong constraints imposed by ribosome function on the mutations in the sequence of ribosomal genes, close microbial species may have identical rDNA sequences, while having clearly different genomes. For example, a comparison between *Cladosporium, Penicillium* and *Fusarium* species at the NCBI Genome and GenBank databases (*Schoch et al., 2012*) will confirm this statement. Such considerations suggested that phylogenetic trees based on sets of genes are potentially more powerful in solving species boundaries than phylogenetic trees based on any single

genes, as the former trees contain information about the simultaneous evolution of various biological processes (*Olmstead & Sweere, 1994*; *Rokas et al., 2003*).

The possibility of using full genomes to create phylogenetic trees becomes more feasible as the number of fully sequenced genomes increases. For example, the full genomic complement of genes/proteins involved in metabolism have been used to reconstruct phylogenies that provide information regarding the evolution of metabolism in various species (*Heymans & Singh, 2003*; *Ma & Zeng, 2004*; *Forst et al., 2006*; *Oh et al., 2006*). This type of genome wide phylogeny reconstruction is impossible for organisms that have not had their genomes fully sequenced and annotated. This is the case for the genus *Diaporthe*, for which the first genome sequencing project started in 2013 (GOLD project Gp0038530) and until now only *Diaporthe* species have their genome sequenced (*Phomopsis longicolla*, *Diaporthe aspalathi*, *Diaporthe ampelina* and *Diaporthe helianthi*) (*Li et al., 2015*; *Baroncelli et al., 2016*; *Li et al., 2016*; *Savitha, Bhargavi & Praveen, 2016*).

Although full genome sequences are still forthcoming for *Diaporthe* species, current species identification and phylogeny reconstruction in the genus are already largely dependent on molecular sequences (*Santos, Correia & Phillips, 2010*). The sequences more frequently used for these studies are: large subunit (LSU) of the ribosomal DNA, intergenic spacers (IGS) of the ribosomal DNA, internal transcribed spacer (ITS) of the ribosomal DNA, translation elongation factor 1-α (*TEF1*) gene, ß-tubulin (*TUB*) gene, histone (*HIS*) gene, calmodulin (*CAL*) gene, actin (*ACT*) gene, DNA-lyase (*APN2*) gene, 60s ribosomal protein L37 (FG1093) gene and mating type genes (MAT-1-1-1 and MAT-1-2-1) (*Farr, Castlebury & Rossman, 2002*; *Farr, Castlebury & Rossman, 2002*; *Castlebury et al., 2003*; *Pecchia, Mercatelli & Vannacci, 2004*; *Schilder et al., 2005*; *Van Rensburg et al., 2006*; *Kanematsu, Adachi & Ito, 2007*; *Santos, Correia & Phillips, 2010*; *Santos et al., 2011*; *Thompson et al., 2011*; *Grasso et al., 2012*; *Sun et al., 2012*; *Udayanga et al., 2012*; *Baumgartner et al., 2013*; *Bienapfl & Balci, 2013*; *Gomes et al., 2013*; *Huang et al., 2013*; *Sun et al., 2013*; *Tan et al., 2013*; *Vidić et al., 2013*; *Gao et al., 2014*; *Udayanga et al., 2014a*; *Udayanga et al., 2014b*; *Wang et al., 2014*).

However, multi-locus phylogenies for the genus *Diaporthe* have only been developed in the last few years (*Schilder et al., 2005*; *Van Rensburg et al., 2006*; *Udayanga et al., 2012*; *Baumgartner et al., 2013*; *Gomes et al., 2013*; *Huang et al., 2013*; *Tan et al., 2013*; *Gao et al., 2014*; *Udayanga et al., 2014a*; *Udayanga et al., 2014b*; *Wang et al., 2014*). In fact, creating phylogenies that include several loci is still possible only for a limited set of species from the genus *Diaporthe*, because not all genes have been sequenced for all tentative species. This is due to many reasons, among which the lack of resources that prevents unlimited sequencing of samples. Nevertheless, a multi-locus approach should always be used for accurate resolution of species in the genus *Diaporthe*.

In recent studies the maximum number of loci used was to create multi loci phylogenies seven (*TEF1*, *TUB*, *HIS*, *CAL*, *ACT*, *APN2* and FG1093), simultaneously sequenced across approximately 80 isolates from 9 *Diaporthe* species (*Udayanga et al., 2014a*). These loci were used to establish the specific limits of *D. eres*. This work provides a good example of how to establish the boundaries for one species within the genus *Diaporthe*. However, if this is

to be extended to the other species of the genus, it is important to determine which loci are the most informative to be sequenced and used in a much wider range of *Diaporthe* species.

With this in mind we asked which combination of frequently sequenced loci better discriminate species boundaries in *Diaporthe*. To answer this question, we considered the ITS, *TEF1*, *TUB*, *HIS* and *CAL* loci, which had been sequenced for 96 different *Diaporthe* species. This paper ranks these loci according to their contribution for improving/decreasing the resolution of *Diaporthe* species determination, as they are added/removed from multi-locus phylogenies analysis.

## MATERIALS & METHODS

### Data collection

In-house PERL scripts were used to search the GenBank and download all sequences from *Diaporthe* and *Phomopsis* species for the 11 loci mentioned in the introduction. We then determined that sequences for ITS, *CAL*, *TUB*, *HIS*, and *TEF1* loci were known in 142 *Diaporthe* and *Phomopsis* isolates, corresponding to 96 different species. Adding any other loci would reduce the number of species. Thus, we have chosen to study these five loci in those 96 species, as a way of maximizing the statistical power of our analysis. Species and gene identifications, as well as, the accession numbers are given in Table S1. The current study used 142 *Diaporthe* isolates that were selected by choosing two isolates per species (whenever they were available), at least one of them being an ex-type isolate. With these constrains in mind, we chose the two isolates for which the sequences were more dissimilar within the same species, in order to maximize intraspecific sequence diversity.

Also considering this intraspecific heterogeneity, we used a larger number of sequence sample for *Diaporthe* species complexes (*Udayanga et al., 2014a*). These are species with a higher than average diversity between individuals. In our case they include *D. sojae*, *D. foeniculacea*, and *D. eres*. For example, the *D. eres* complex includes strains CBS 113470, CBS 116953, CBS 200.39, and CBS 338.89, some of which were originally classified as *D. nobilis* and later reclassified into the *D. eres* complex (*Gomes et al., 2013*; *Udayanga et al., 2014a*). In addition, we used more than one ex-type isolate for the species complexes, because these species are highly heterogeneous. All sequence data used in this study have been validated and published previously (*Castlebury et al., 2003*; *Van Niekerk et al., 2005*; *Santos et al., 2011*; *Gomes et al., 2013*; *Udayanga et al., 2014a*).

As species concept we used the criteria of Genealogical Concordance Phylogenetic Species Recognition (GCPSR) to resolve species boundaries based on individual and combined analyses of the five genes.

### Sequence alignment and phylogenetic analyses

Five multiple alignments, one per locus, were created using the software ClustalX2.1 (*Larkin et al., 2007*), and the following parameters: pairwise alignment parameters (gap opening = 10, gap extension = 0.1) and multiple alignment parameters (gap opening = 10, gap extension = 0.2, transition weight = 0.5, delay divergent sequences = 25%), and optimized manually with BioEdit (*Hall, 1999*). The alignments for the individual locus were then concatenated into all possible combinations of two, three, four and five loci. This

generated 31 alternatives multiple alignments, counting the five multiple alignments for the individual genes and the alignment for the five concatenated gene sequences. MEGA6 (*Tamura et al., 2013*) was used to create and analyse phylogenetic trees for each of the 31 alignments, independently using two alternative methods (Maximum Parsimony (MP) and Maximum Likelihood (ML); *Li, 1997*). MEGA6 was also used to determine the best evolution models to be used for building the ML tree from each multiple alignment, as described previously (*Tamura et al., 2013*). These models are listed in Table 1. Each tree was bootstrapped 1,000 times, and branches that split in less than 90% of the 1,000 trees were condensed. MP trees were obtained using the Tree-Bisection-Reconnection (TBR) algorithm (*Nei & Kumar, 2000*) with search level 1, in which the initial trees were obtained by the random addition of sequences (10 replicates). The initial trees for the heuristic ML search were obtained by applying the Neighbor-Joining method to a matrix of pairwise distances estimated using the Maximum Composite Likelihood (MCL) approach, allowing for some sites to be evolutionarily invariable ([+I], 0.0000% sites). As in *Gomes et al. (2013)*, we choose *Diaporthella corylina* (CBS 121124) as outgroup.

## Comparing trees
### Tree scores
MEGA6 was used to create and analyse all MP and ML phylogenetic trees. As a first approximation, we compare the likelihood values between ML trees and the MP scores between MP trees (Tables 2 and 3) for identifying the best and worst trees of each type.

The length of an MP tree estimates phylogenetic tree resolution. This value is also dependent on the length of the sequences that are used to build the tree. This means that comparing tree lengths for trees built using a varying number of loci should also consider normalizing the length of the tree by the corresponding size of the aligned sequence (Table 2). This normalization allows us to estimate which loci provide more added value when it comes to species resolution.

ML tree building methods seek the tree that is more likely (the highest likelihood), based on a probabilistic model of sequence evolution. The best ML tree has the lowest—log likelihood scores and worst ML tree has the highest—log likelihood value. This likelihood is also dependent on the length of the alignment. In order to be able to compare all the trees among them we also normalized the values of—log Likelihood in the same way of the MP length (Table 3). This means that comparing tree log likelihoods for trees built using a varying number of loci should also consider normalizing the log likelihood of the tree by the corresponding size of the aligned sequence (Table 3).

### Tree distances
All trees we build have the same species. Thus, we are able to measure the difference between every possible pair of trees, based on the analysis of the symmetric distance between equal leafs in two trees (*Robinson & Foulds, 1981*). This distance was calculated for all pairs of MP trees using the Treedist methods of the PHYLIP suite of programs (*Felsenstein, 1989*). The same calculations were made for all pairs of ML trees. For these calculations we used condensed trees with a 90% bootstrap cut-off value. This allows us to measure how
**Table 1** Models used to construct the ML trees.

| Tree | Model | References |
|------|-------|-----------|
| ITS | Tamura-Nei | *Tamura & Nei (1993)* |
| TEF1 | Hasegawa-Kishino-Yano | *Hasegawa, Kishino & Yano (1985)* |
| TUB | Hasegawa-Kishino-Yano | *Hasegawa, Kishino & Yano (1985)* |
| HIS | General Time Reversible | *Nei & Kumar (2000)* |
| CAL | Tamura 3-parameter | *Tamura (1992)* |
| ITS-TEF1 | Tamura-Nei | *Tamura & Nei (1993)* |
| ITS-TUB | Tamura-Nei | *Tamura & Nei (1993)* |
| ITS-HIS | Tamura-Nei | *Tamura & Nei (1993)* |
| ITS-CAL | Tamura-Nei | *Tamura & Nei (1993)* |
| TEF1-TUB | Hasegawa-Kishino-Yano | *Hasegawa, Kishino & Yano (1985)* |
| TEF1-HIS | Tamura-Nei | *Tamura & Nei (1993)* |
| TEF1-CAL | Hasegawa-Kishino-Yano | *Hasegawa, Kishino & Yano (1985)* |
| TUB-HIS | General Time Reversible | *Nei & Kumar (2000)* |
| TUB-CAL | Hasegawa-Kishino-Yano | *Hasegawa, Kishino & Yano (1985)* |
| HIS-CAL | Hasegawa-Kishino-Yano | *Hasegawa, Kishino & Yano (1985)* |
| ITS-TEF1-TUB | Tamura-Nei | *Tamura & Nei (1993)* |
| ITS-TEF1-HIS | General Time Reversible | *Nei & Kumar (2000)* |
| ITS-TEF1-CAL | Tamura-Nei | *Tamura & Nei (1993)* |
| ITS-TUB -HIS | General Time Reversible | *Nei & Kumar (2000)* |
| ITS-TUB-CAL | Tamura-Nei | *Tamura & Nei (1993)* |
| ITS-HIS-CAL | Tamura-Nei | *Tamura & Nei (1993)* |
| TEF1-TUB-HIS | General Time Reversible | *Nei & Kumar (2000)* |
| TEF1-TUB-CAL | Hasegawa-Kishino-Yano | *Hasegawa, Kishino & Yano (1985)* |
| TEF1-HIS-CAL | Tamura-Nei | *Tamura & Nei (1993)* |
| TUB-HIS-CAL | Hasegawa-Kishino-Yano | *Hasegawa, Kishino & Yano (1985)* |
| ITS-TEF1-TUB-HIS | General Time Reversible | *Nei & Kumar (2000)* |
| ITS-TEF1-TUB-CAL | Tamura-Nei | *Tamura & Nei (1993)* |
| ITS-TEF1-HIS-CAL | Tamura-Nei | *Tamura & Nei (1993)* |
| ITS-TUB-HIS-CAL | Tamura-Nei | *Tamura & Nei (1993)* |
| TEF1-TUB-HIS-CAL | Hasegawa-Kishino-Yano | *Hasegawa, Kishino & Yano (1985)* |
| ITS-TEF1-TUB-HIS-CAL | General time reversible | *Nei & Kumar (2000)* |

adding/removing a locus to/from the multiple alignments causes the resulting phylogenetic tree to change.

### *Testing Phylogenetic informativeness and identification of species boundaries*

We used PhyDesign (*López-Giráldez & Townsend, 2011*) to establish the informativeness of the various combinations of loci alignments, as described in *Udayanga et al. (2014a)*. We also manually analyzed all trees to identify all cases where isolates of the same species did not cluster together. This allowed us to determine the loci that provided the best species resolution.

**Table 2  MP trees scores.**

| Tree | No. trees | Length | Normalized length | Consistency index | Retention index | Composite index | Parsimony-informative sites |
|------|-----------|--------|-------------------|-------------------|-----------------|-----------------|------------------------------|
| *1gene* | | | | | | | |
| ITS | 1 | 1,200 | 1.970 | 0.278906 | 0.765634 | 0.244365 | 0.213540 |
| TEF1 | 1 | 2,830 | 4.647 | 0.280810 | 0.773915 | 0.229713 | 0.217323 |
| TUB | 1 | 1,628 | 2.673 | 0.349176 | 0.785012 | 0.289798 | 0.274107 |
| HIS | 1 | 1,880 | 3.087 | 0.285557 | 0.729297 | 0.224608 | 0.208256 |
| CAL | 1 | 2,234 | 3.668 | 0.355136 | 0.816321 | 0.304750 | 0.289905 |
| *2 genes* | | | | | | | |
| ITS-TEF1 | 1 | 4,218 | 6.926 | 0.267368 | 0.756266 | 0.219278 | 0.202201 |
| ITS-TUB | 1 | 2,977 | 4.888 | 0.303147 | 0.758804 | 0.250811 | 0.230029 |
| ITS-HIS | 1 | 3,268 | 5.366 | 0.266073 | 0.721901 | 0.212506 | 0.192078 |
| ITS-CAL | 1 | 3,657 | 6.005 | 0.308194 | 0.780338 | 0.259686 | 0.240496 |
| TEF1-TUB | 2 | 4,535 | 7.447 | 0.300317 | 0.772148 | 0.245351 | 0.231889 |
| TEF1-HIS | 1 | 4,828 | 7.928 | 0.275606 | 0.749486 | 0.220282 | 0.206562 |
| TEF1-CAL | 1 | 5,206 | 8.548 | 0.304724 | 0.784949 | 0.252402 | 0.239193 |
| TUB-HIS | 1 | 3,606 | 5.921 | 0.306263 | 0.746843 | 0.244391 | 0.228730 |
| TUB-CAL | 1 | 3,975 | 6.527 | 0.342310 | 0.795145 | 0.287052 | 0.272186 |
| HIS-CAL | 1 | 4,267 | 7.007 | 0.311460 | 0.770357 | 0.255101 | 0.239935 |
| *3 genes* | | | | | | | |
| ITS-TEF1-TUB | 1 | 5,942 | 9.757 | 0.285318 | 0.758687 | 0.232892 | 0.216467 |
| ITS-TEF1-HIS | 3 | 6,233 | 10.235 | 0.266876 | 0.740786 | 0.214166 | 0.197698 |
| ITS-TEF1-CAL | 1 | 6,609 | 10.852 | 0.290524 | 0.771371 | 0.240083 | 0.224102 |
| ITS-TUB-HIS | 1 | 4,989 | 8.192 | 0.288178 | 0.737853 | 0.231161 | 0.212633 |
| ITS-TUB-CAL | 1 | 5,378 | 8.831 | 0.315121 | 0.775889 | 0.262285 | 0.244499 |
| ITS-HIS-CAL | 2 | 5,661 | 9.296 | 0.293677 | 0.757132 | 0.240207 | 0.222352 |
| TEF1-TUB-CAL | 1 | 6,910 | 11.346 | 0.311702 | 0.781378 | 0.257255 | 0.243557 |
| TEF1-TUB-HIS | 1 | 6,537 | 10.734 | 0.290338 | 0.754311 | 0.233090 | 0.219005 |
| TEF1-HIS-CAL | 1 | 7,209 | 11.837 | 0.294419 | 0.766557 | 0.239569 | 0.225689 |
| TUB-HIS-CAL | 1 | 5,962 | 9.790 | 0.318135 | 0.770532 | 0.260290 | 0.245133 |
| *4 genes* | | | | | | | |
| ITS-TEF1-TUB-HIS | 1 | 7,934 | 13.028 | 0.281222 | 0.747108 | 0.226279 | 0.210103 |
| ITS-TEF1-TUB-CAL | 1 | 8,326 | 13.672 | 0.298775 | 0.770405 | 0.245945 | 0.230178 |
| ITS-TEF1-HIS-CAL | 1 | 8,622 | 14.158 | 0.284827 | 0.757809 | 0.231684 | 0.215844 |
| ITS-TUB-HIS-CAL | 1 | 7,364 | 12.092 | 0.302877 | 0.759944 | 0.247364 | 0.230170 |
| TEF1-TUB-HIS-CAL | 2 | 8,911 | 14.632 | 0.301867 | 0.767101 | 0.245686 | 0.231563 |
| *5 genes* | | | | | | | |
| ITS-TEF1-TUB-HIS-CAL | 1 | 10,327 | 16.957 | 0.292768 | 0.759604 | 0.238098 | 0.222388 |
**Table 3** Data from the likelihood values using ML trees.

| Tree | −log Likelihood | Normalizad −log likelihood |
|---|---|---|
| *1 gene* | | |
| ITS | −6778.9324 | −11.1313 |
| TEF1 | −12771.9747 | −20.9720 |
| TUB | −8921.3230 | −14.6491 |
| HIS | −9330.7481 | −15.3214 |
| CAL | −11.756.6407 | −19.3048 |
| *2 genes* | | |
| ITS-TEF1 | −20494.0008 | −33.6519 |
| ITS-TUB | −16381.1047 | −26.8984 |
| ITS-HIS | −16835.4062 | −27.6443 |
| ITS-CAL | −19449.5000 | −31.9368 |
| TEF1-TUB | −22209.9657 | −36.4696 |
| TEF1-HIS | −22707.1478 | −37.2860 |
| TEF1-CAL | −25263.7157 | −41.4839 |
| TUB-HIS | −18720.0479 | −30.7390 |
| TUB-CAL | −21286.5020 | −34.9532 |
| HIS-CAL | −21896.7086 | −35.9552 |
| *3 genes* | | |
| ITS-TEF1-TUB | −29959.8491 | −49.1952 |
| ITS-TEF1-HIS | −30409.1656 | −49.9329 |
| ITS-TEF1-CAL | −33105.3032 | −54.3601 |
| ITS-TUB-HIS | −26256.8160 | −43.1146 |
| ITS-TUB-CAL | −29008.0228 | −47.6322 |
| ITS-HIS-CAL | −29425.9498 | −48.3185 |
| TEF1-TUB-CAL | −34699.3754 | −56.9776 |
| TEF1-TUB-HIS | −32201.5900 | −52.8762 |
| TEF1-HIS-CAL | −35160.1260 | −57.7342 |
| TUB-HIS-CAL | −31194.9713 | −51.2233 |
| *4 genes* | | |
| ITS-TEF1-TUB-HIS | −39950.3574 | −65.5999 |
| ITS-TEF1-TUB-CAL | −42574.6960 | −69.9092 |
| ITS-TEF1-HIS-CAL | −42940.9069 | −70.5105 |
| ITS-TUB-HIS-CAL | −38862.9726 | −63.8144 |
| TEF1-TUB-HIS-CAL | −44608.0234 | −73.2480 |
| *5 genes* | | |
| ITS-TEF1-TUB-HIS-CAL | −52626.8115 | −86.4151 |

## RESULTS

We analyse 142 isolates from 96 *Diaporthe* species for which the ITS, *CAL*, *TUB*, *HIS*, and *TEF1* loci had been sequenced (Table S1). The alignments for each locus were then concatenated in all possible 31 combinations of one, two three, four and five genes. Alignment characteristics for this study are reported in Table 4. Each combination was used to build a ML and ME phylogenetic trees. Each tree was bootstrapped 1,000 times and every tree used is a condensed tree with a 90% cut-off. Alignments and trees were deposited in TreeBase (Study Accession: S20343).

### Best and worst resolving phylogenetic trees

The "quality" (resolution) of the individual phylogenetic trees was determined as described in methods.

Figures 1 and 2 present the condensed MP and ML trees build from the concatenated multiple alignments of the five loci, respectively. Phylograms showing all complete trees are given as (Figs. S1 and S2, respectively). These trees are the best resolving trees built for each method, as indicated by the scores shown in Table 2 for MP trees and in Table 3 for ML trees.

The increase in tree length (Table 2) and log-likelihood scores (Table 3) of the trees with the increase in number of loci indicates that resolution of the trees is directly correlated with the number of loci used to build them. This is also true for the tree scores and log-likelihood scores normalized by alignment length. Thus, the worst trees are built using the multiple alignments for only one locus. Within the one-locus trees, the best MP (Fig. 3 and Fig. S3) and ML (Fig. 4 and Fig. S4) condensed trees are shown in Figs. 3 and 4. *TEF1* trees have the highest values for length and-log likelihood.

### Choosing the most informative loci for sequencing

The previous results indicate that, whenever possible, all five loci should be sequenced, in order to better differentiate between *Diaporthe* species. However, this might not always be possible. In situations where only a subset of one, two, three, or four out of the five loci can be sequenced, which sequences might be more informative? This can be roughly answered in two steps.

The first step is done by measuring how adding/removing a locus to/from the multiple alignments causes the resulting phylogenetic tree to change. These changes can be measured by calculating the symmetric distance between the two trees and by analysing if species resolution changes when the relevant locus is added or removed. The smaller the changes are, the less informative the locus is. The symmetric distance matrices between every pair of MP (Table S2) or ML (Table S3 ) trees were calculated as described in methods. Table 5 summarize these results and show how many changes are observed on average when a specific locus is removed from a multi-locus tree. On average, the ITS locus is the least informative one, closely followed by the HIS locus. The third locus whose removal causes the least changes in the trees is CAL. This is true for both, the MP and the ML trees.

The second step is done by evaluating the changes in the resolution of the trees when a locus is removed from the multiple alignments. A more detailed analysis of Tables 3–5

**Table 4 Alignments characteristics.**

| Locus | No. Characters | No. Conserved sites (in %) | No. variable sites (in %) | No. Parsim-info sites (in %) |
|---|---|---|---|---|
| *1 gene* | | | | |
| ITS | 609 | 350 (57) | 235 (39) | 177 (29) |
| TEF1 | 535 | 128 (24) | 382 (71) | 328 (61) |
| TUB | 603 | 220 (36) | 323 (54) | 279 (46) |
| HIS | 688 | 329 (48) | 311 (45) | 259 (38) |
| CAL | 667 | 194 (29) | 425 (64) | 370 (55) |
| *2 genes* | | | | |
| ITS-TEF1 | 1,149 | 478 (42) | 617 (54) | 505 (44) |
| ITS-TUB | 1,217 | 570 (47) | 558 (46) | 456 (37) |
| ITS-HIS | 1,302 | 679 (52) | 546 (42) | 436 (33) |
| ITS-CAL | 1,281 | 544 (42) | 660 (52) | 547 (43) |
| TEF1-TUB | 1,143 | 348 (30) | 705 (62) | 607 (53) |
| TEF1-HIS | 1,228 | 457 (37) | 693 (56) | 587 (48) |
| TEF1-CAL | 1,207 | 322 (27) | 807 (67) | 698 (58) |
| TUB-HIS | 1,296 | 549 (42) | 634 (49) | 538 (42) |
| TUB-CAL | 1,275 | 414 (32) | 748 (59) | 649 (51) |
| HIS-CAL | 1,360 | 523 (38) | 736 (54) | 629 (46) |
| *3 genes* | | | | |
| ITS-TEF1-TUB | 1,757 | 698 (40) | 940 (54) | 784 (45) |
| ITS-TEF1-HIS | 1,842 | 807 (44) | 928 (50) | 764 (41) |
| ITS-TEF1-CAL | 1,821 | 672 (37) | 1,042 (57) | 875 (48) |
| ITS-TUB-HIS | 1,910 | 899 (47) | 869 (45) | 715 (37) |
| ITS-TUB-CAL | 1,889 | 764 (40) | 983 (52) | 826 (44) |
| ITS-HIS-CAL | 1,974 | 873 (44) | 971 (49) | 806 (41) |
| TEF1-TUB-CAL | 1,815 | 542 (30) | 1,130 (62) | 977 (54) |
| TEF1-TUB-HIS | 1,836 | 677 (379) | 1,016 (55) | 866 (47) |
| TEF1-HIS-CAL | 1,900 | 651 (34) | 1,118 (59) | 957 (50) |
| TUB-HIS-CAL | 1,968 | 743 (38) | 1,059 (54) | 908 (46) |
| *4 genes* | | | | |
| ITS-TEF1-TUB-HIS | 2,450 | 1,027 (42) | 1,251 (51) | 1,043 (43) |
| ITS-TEF1-TUB-CAL | 2,429 | 892 (37) | 1,365 (56) | 1,154 (48) |
| ITS-TEF1-HIS-CAL | 2,514 | 1,001 (40) | 1,353 (54) | 1,134 (45) |
| ITS-TUB-HIS-CAL | 2,582 | 1,093 (42) | 1,294 (50) | 1,085 (42) |
| TEF1-TUB-HIS-CAL | 2,508 | 871 (35) | 1,441 (57) | 1,236 (49) |
| *5 genes* | | | | |
| ITS-TEF1-TUB-HIS-CAL | 3,102 | 1,221 (39) | 1,676 (54) | 1,413 (46) |

reveals that removing the ITS locus from any MP or ML multi-loci tree causes the smallest decrease in MP tree length and in ML tree likelihood. Hence, if only four loci can be sequenced these should be *TEF1-TUB-CAL-HIS*. The second locus with the least effect in tree resolution is *TUB*, closely followed by *HIS*. Given that, as measured in step one of the process, average differences between trees when *HIS* is removed are much smaller than differences between trees when *TUB* is removed, if only three loci can be sequenced

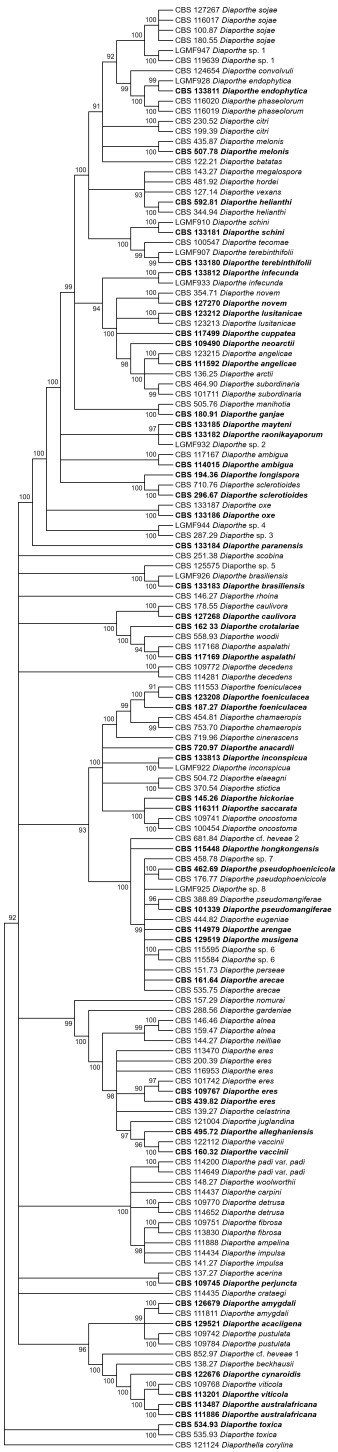

**Figure 1 MP condensed tree with a 90% cut-off, build using the five loci *TEF1-TUB-CAL-HIS*-ITS for the 96 *Diaporthe* species.** Ex-type or ex-epitype or isotype isolates are represented in bold.

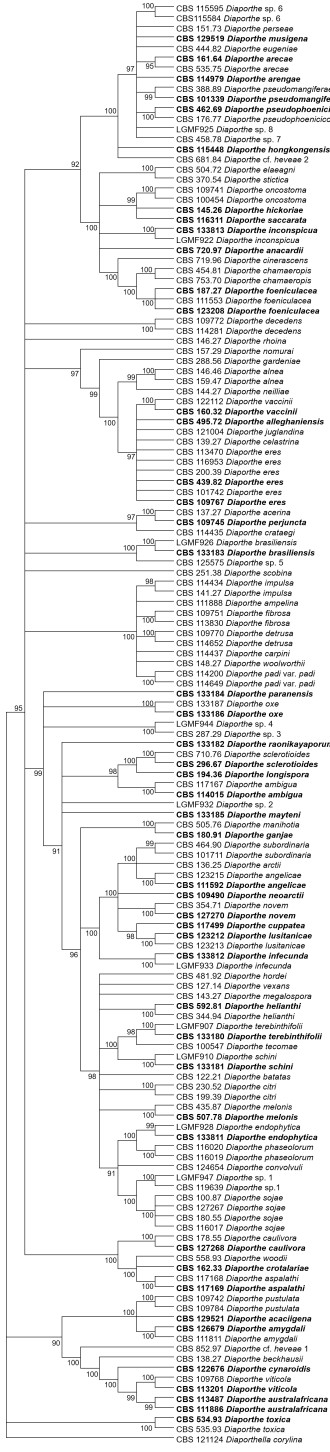

**Figure 2  ML condensed tree with a 90% cut-off, build using the five loci *TEF1-TUB-CAL-HIS*-ITS for the 96 *Diaporthe* species.** The percentage of trees in which the associated taxa clustered together is shown next to the branches. Ex-type, ex-epitype, or isotype isolates are represented in bold.

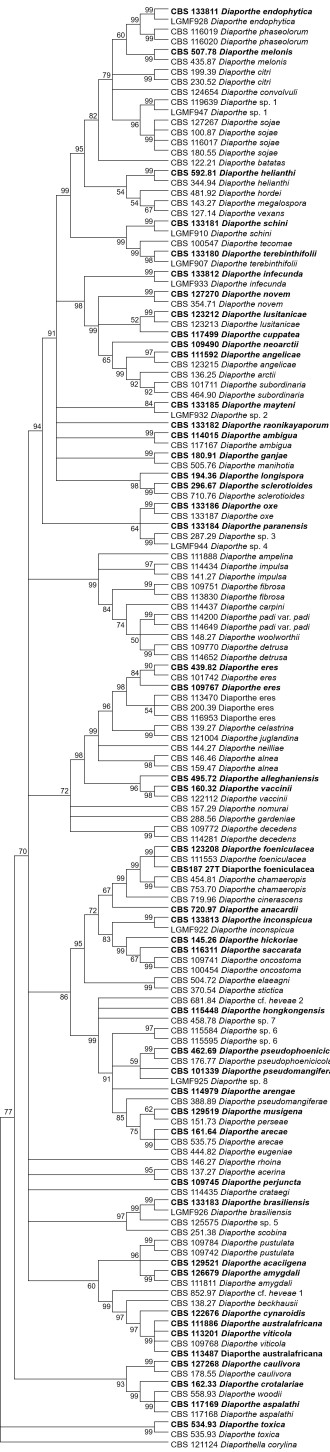

**Figure 3** **MP condensed tree with a 90% cut-off build using the *TEF1* locus for the 96 *Diaporthe*
species.** This locus generates the best single locus trees for the MP method. Ex-type, ex-epitype, or isotype
isolates are represented in bold.

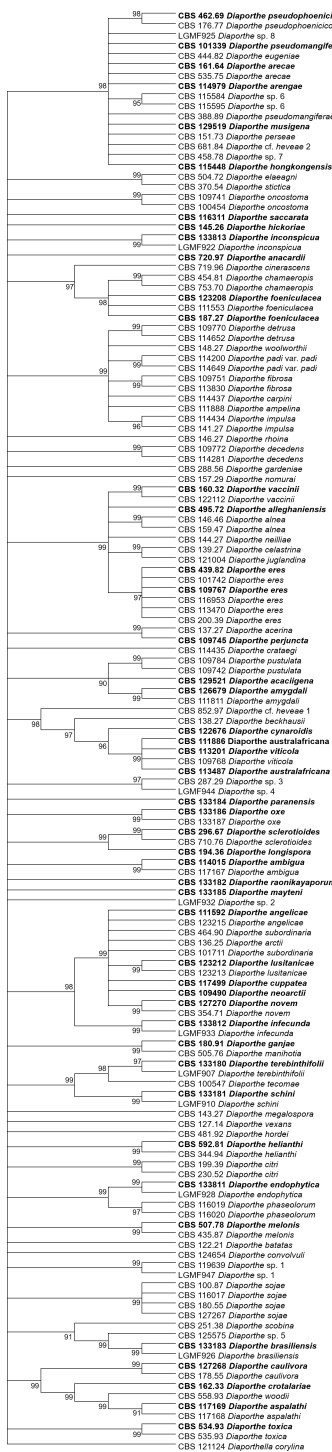

**Figure 4** **ML condensed tree with a 90% cut-off, build using the *TEF1* locus for the 96 *Diaporthe* species.** This locus generates the best single locus trees for the ML method. The percentage of trees in which the associated taxa clustered together is shown next to the branches. Ex-type, ex-epitype, or isotype isolates are represented in bold.

**Table 5 Average changes in tree resolution when a locus is added or removed.** Each row indicates the locus that is added to the trees. Each column indicates the difference between trees build using $n$ or $n-1$ loci. For example, row ITS, columns $4 \to 3$, indicate the average differences between every pair of 3- and 4-loci trees that include the ITS locus, using either a MP or a ML approach. The higher the number, the more different the two trees in the pair are, on average. "Average" columns indicate the average changes for all columns when a specific locus is considered. Darker cells indicate smaller average changes (and thus smaller information losses) when a locus is added from phylogenetic trees.

| | MP | | | | | | ML | | | | |
|---|---|---|---|---|---|---|---|---|---|---|---|
| | $5 \to 4$ | $4 \to 3$ | $3 \to 2$ | $2 \to 1$ | Overall | | $5 \to 4$ | $4 \to 3$ | $3 \to 2$ | $2 \to 1$ | Overall |
| ITS | 1.416 | 1.407 | 31.9 | 36.875 | 17.8995 | ITS | 12 | 23.75 | 24.3888889 | 31.75 | 22.9722222 |
| TEF1 | 2.963 | 2.95075 | 44.7333333 | 44.375 | 23.7555208 | TEF1 | 22 | 25.25 | 30.6111111 | 42.25 | 30.0277778 |
| TUB | 1.705 | 1.70575 | 41.4 | 43.75 | 22.1401875 | TUB | 16 | 24.375 | 28.6666667 | 41.375 | 27.6041667 |
| HIS | 2.001 | 1.998 | 41.2 | 40.5 | 21.42475 | HIS | 12 | 19 | 24.8888889 | 37 | 23.2222222 |
| CAL | 2.393 | 29 | 42.6 | 42.125 | 29.0295 | CAL | 10 | 24.5 | 25.3333333 | 39.125 | 24.7395833 |

these should be *TEF1-TUB-CAL*. If only two loci can be sequenced, we suggest *TEF1-TUB*, as removing *CAL* has the least average effect on trees. Finally, if only one locus can be sequenced tree resolution suggests that this locus should be *TEF1*. *TEF1* trees are the best single locus MP and ML trees (Figs. 3 and 4).

## Phylogenetic informativeness and identification of species boundaries

Figure 5 shows that the *TEF1* sequence is the most informative for species separation, both globally and per alignment site. In addition, we also see that the ITS sequence is the least informative to resolve *Diaporthe* species (Fig. 5). The five loci can be ranked from most to least informative for *Diaporthe* species separation as follows: *TEF1 > HIS > CAL > TUB > ITS*.

The dataset we used for this analysis is as close as we currently can get to a standard set of well separated *Diaporthe* species, taking into account that the five loci we analyse needed to be sequenced for all individuals in the set. Taking this into account, an inspection of the trees is required to understand, on top of all the statistical analyses, if species are well separated or not.

We see that, in general, the addition of a new locus to the alignment decreases the number of isolates from the same species that do not cluster together (separation errors). Therefore, the tree of 5 loci has less separation errors than 4-loci trees, which in turn have less separation errors than the 3-loci trees, and so on. As expected from our previous analysis, the *TEF1* tree provides the best single locus ML tree, *TEF1-TUB* tree provides the best 2-loci ML tree, *TEF1-TUB-CAL* the best 3-locus ML tree. The results from the MP trees are qualitatively similar although, in general, these trees have more separation errors that the ML ones.

## DISCUSSION

Identifying species boundaries in organisms is a difficult task, as theoretical and practical definitions of species are not always consistent with each other (*Doolittle & Zhaxybayeva, 2009*; *Giraud et al., 2008*). While *Woese & Fox (1977)* suggested using ribosomal sequences to define species borders, such sequences are not always the best choice. For example,

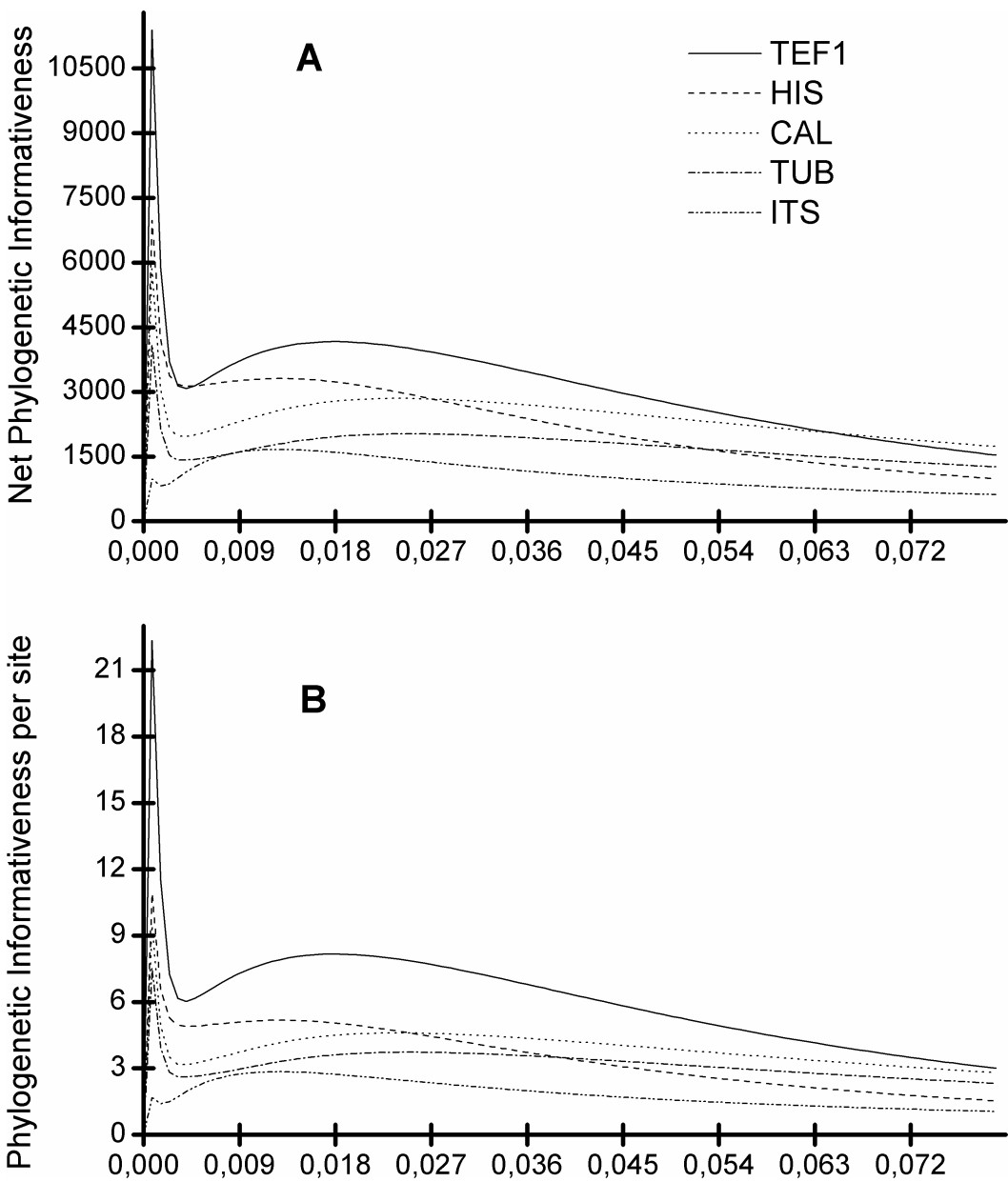

**Figure 5** **Profiles of phylogenetic informativeness for the 96 *Diaporthe* species and 5 loci.** (A) Net Phylogenetic informativeness. (B) Phylogenetic informativeness per site.

searching GenBank will reveal that some *Cladosporium, Penicillium* and *Fusarium* species cannot be differentiated using ITS (*Schoch et al., 2012*).

More recent work suggests that trees based on multi-loci sequence analysis (MLSA) provide more accurate estimations of phylogeny than single gene trees, if appropriate loci are used (*Gadagkar, Rosenberg & Kumar, 2005*; *Mirarab, Bayzid & Warnow, 2016*). Briefly, MLSA concatenates sequence alignments from multiple genes and uses the concatenated sequences to determine phylogenetic relationships. This method appears to more optimally resolve the phylogenetic position of species in the same or in closely related genera (*Hanage,*

*Fraser & Spratt, 2006*). An increase in the number of loci used to build MLSA phylogenetic trees positively correlates to sensitivity and accuracy in species separation (*Rokas et al., 2003*; *Udayanga et al., 2011*). In contrast, increasing the number of species in the alignment leads to a decrease in the ability to separate them accurately, unless a higher number of appropriate loci are used to maintain the quality of that separation (*Bininda-Emonds et al., 2001*; *Kim, 1998*; *Poe & Swofford, 1999*; *Rokas et al., 2003*; *Udayanga et al., 2011*). The choice of appropriate loci to be used in such trees can be optimized in genera with a large number of sequenced genomes, because in such cases it is possible to make full genome studies to identify the best set of loci to separate species. Nevertheless, the amount of information that must be analysed for doing so could become prohibitive (*Thangaduras & Sangeetha, 2013*).

The choice of appropriate loci that optimizes species separation is harder when fully sequenced genomes are not available, as is the case for the genus *Diaporthe*. Nevertheless, MLSA phylogenetic studies of *Diaporthe* species have been done using loci that have been chosen in a more or less *ad hoc* manner, by taking into account how conserved they were in different fungal genus (*Baumgartner et al., 2013*; *Gao et al., 2014*; *Gomes et al., 2013*; *Huang et al., 2013*; *Schilder et al., 2005*; *Tan et al., 2013*; *Udayanga et al., 2012*; *Udayanga et al., 2014a*; *Udayanga et al., 2014b*; *Van Rensburg et al., 2006*; *Wang et al., 2014*). In general, these studies show that MLSA phylogenetic trees provide higher resolution for *Diaporthe* species than single locus phylogenetic trees (*Huang et al., 2013*; *Udayanga et al., 2012*; *Van Rensburg et al., 2006*).

The current study addresses the problem of which loci are best for accurate species separation in the genus *Diaporthe* in a systematic manner. *Walker et al. (2012)* performed a similar study. While we use five non-coding loci to study species separation in *Diaporthe*, those authors employed two single copy protein-coding genes (FG1093 and MS204) to study species separation in Sordariomycetes. While *Walker et al. (2012)* analysed various aspects of codon conservation and substitution rates, these analyses are meaningless for our sequence dataset. The use of non-coding sequences is favoured in *Diaporthe* species separation because coding sequences are typically too conserved to allow for appropriate separation within the genus.

The major contributions of this paper are two-fold. First, our work confirms that the quality of species separation in phylogenetic trees increases with the number of loci used to build phylogenetic trees. Second and more importantly it identifies the best combination of loci that one should use for building those phylogenetic trees, if only one, two, three, or four loci can be sequenced. To achieve this, we took the most commonly sequenced loci for 142 *Diaporthe* isolates and studied which loci optimize species differentiation in the genus. We chose only loci that are commonly sequenced for members of the genus. Then, we selected a sequence dataset that was experimentally validated by others (*Castlebury et al., 2003*; *Van Niekerk et al., 2005*; *Santos et al., 2011*; *Gomes et al., 2013*) before being deposited in GenBank. Whenever possible we favoured sequences from ex-type isolates and produced via low throughput, high fidelity, sequencing methods. In addition, our sequence selection maximized intraspecific sequence variation, which in turn maximizes the possibility that intra-specific hyperdiversity could be higher than interspecific diversity.

Thus, species separation through phylogenetic trees in our sample is made more difficult by our sequence selection, making our analysis more robust. In this paper we only show and analyse condensed MP and ML trees, using a cut-off of 90%, which means that our trees are very robust to gene order, as a significant amount of bootstrapping was used to calculate them. In fact, to test that, we performed a side experiment where we changed the order of the locus sequences in the alignments and recalculated the trees (Fig. S5).

We found that species differentiation is optimized by creating phylogenetic trees built from the multiple sequence alignment of five loci: *TEF1-TUB-HIS-CAL*-ITS. However, little information is lost when ITS locus is removed and only the other four loci are used to simultaneously build the phylogeny. In addition, we also provide researchers with a ranking of best loci to sequence if only one, two, three or four of the loci can be sequenced.

It may be surprising that the ribosomal ITS locus is the least informative of the five loci when it comes to separating *Diaporthe* species. However, *Santos, Correia & Phillips (2010)* found that the ITS region in *Diaporthe* is evolving at much faster rates than *TEF1* or even *MAT* genes. Hence, what seems to be happening is that ITS sequences present a wider variation than is advisable for creating precise species boundaries. Therefore a slowly evolving gene region should be utilized in order to establish precise species limits (*Udayanga et al., 2012*).

DNA barcoding (*Kress et al., 2014*) refers to the use of standard short gene sequences to identify species. The use of DNA barcoding implies that an effort should be made to standardize the use of the loci for phylogenetic studies. ITS is the official DNA barcode region in fungi (*Schoch et al., 2012*). This work supports previous studies whose results suggest that using ITS as a standard for species separation in fungi should be discontinued (*Gomes et al., 2013*; *Thangaduras & Sangeetha, 2013*). Our results strongly recommend that TEF1 should be used instead, at least in the genus *Diaporthe*. This is consistent with and further develops previously published results, which proposed either *TEF1*, *HIS*, or *APN2* as alternative locus for barcoding in the genus (*Santos, Correia & Phillips, 2010*; *Udayanga et al., 2014b*). However, *Gomes et al. (2013)*, using Bayesian analysis, consider *HIS* and *TUB* as best resolving genes. Nevertheless, considering that *Gomes et al. (2013)* use shorter sequences than those used here, one is tempted to cautiously analyse and reinterpret their conclusions.

Despite the *TEF* tree appears to be a better species separator than the 5-loci tree, the true is that, the alignment used to build the 5-loci tree is roughly five times larger than that for the *TEF* tree. This means that, with a larger number of positions, there is bound to be more variability in the bootstrapping of the 5-loci tree than in the bootstrapping of the *TEF* tree. Hence, the observation that the *TEF* give better resolution than 5 loci results from a statistical artefact. This fact occurs when focusing on the *D. eres* complex clade. For example, in the case of the *D. eres* complex, all the species are grouped in the same clade in both cases (*D. alleghaniensis*, *D. alnea*, *D. celastrina*, *D. bicincta*, *D. eres*, *D. neilliae* and *D. vaccinii*). However, in the 5-loci trees the resolution of this species complex is better. This is especially important as phylogenetic analyses of the *D. eres* complex often revealed ambiguous clades with short branch and moderate statistic supports due to their high variability. *Udayanga et al. (2014a)* studied this problematic by using different genes,

whose sequences are not available for the other *Diaporthe* species we consider. Therefore, we could not incorporate their data in our study. We also note that one possible explanation for the observation that some species of the *D. eres* complex do not "group" in the same clade could be due to the fact that they are not really *D. eres*. However, to test that, we would need to actually obtain samples of the complex, re-sequence and analyse them in order to clarify the species boundaries in this group.

The problem of species boundary identification is very relevant in the genus *Diaporthe*, where a general taxonomic revision based on molecular analysis is probably overdue. Such a revision could then be used to improve the annotation of sequences in public databases, such as GenBank. For example, many of the sequences we use in our analysis are still assigned to species that have already been reclassified. This also emphasizes that a standard procedure with minimal information required for submitting new *Diaporthe* species needs to be put in place in order to avoid unnecessary creation of new species (*Udayanga et al., 2014b*). Furthermore, as also suggested by *Gomes et al. (2013)* we feel that this revision should be made using molecular data. Any new *Diaporthe* species report should be accompanied by molecular data that supports the identification of the individual as a new species. In addition, we feel that a proper taxonomic revision of the genus should also consider morphological descriptions and epitypification of species as previously suggested (*Gomes et al., 2013*; *Udayanga et al., 2014b*).

## CONCLUSIONS

Our results indicate that:

- In order of effectiveness the best sets of loci for resolving *Diaporthe* species are *TEF1-TUB-CAL-HIS*-ITS, *TEF1-TUB-CAL-HIS*, *TEF1-TUB-CA* L, *TEF1-TUB* and *TEF1*.
- The *TEF1* locus is a better candidate for single locus DNA barcoding in the genus *Diaporthe* than the ITS locus.
- Multi-loci DNA barcoding will provide a more accurate species separation in the genus than single locus barcoding. Furthermore, a four loci barcoding including *TEF1-TUB-HIS-CAL* will be almost as effective as a five loci barcoding including ITS-*TEF1-TUB-HIS-CAL*.

## ACKNOWLEDGEMENTS

We thank Anabel Usié and R Benfeitas for assistance with the creation of the Perl scripts.

### Funding

This work was financed by European Funds through COMPETE and by National Funds through the Portuguese Foundation for Science and Technology (FCT) within project PANDORA (PTDC/AGR-FOR/3807/2012 –FCOMP-01-0124-FEDER-027979). The authors received financing from FCT to CESAM (UID/AMB/50017/2013), Artur

Alves (FCT Investigator Programme –IF/00835/2013) and a post-doctoral grant to Liliana Santos (SFRH/BPD/90684/2012), grants BFU2010-17704 from Ministerio de Ciencia e Innovación, 2009SGR809 from Generalitat de Catalunya, and bridge grants from the Dean for Research and the Departament de Ciències Mèdiques Bàsiques of the University of Lleida (Spain) to Rui Alves. There was no additional external funding received for this study. The funders had no role in study design, data collection and analysis, decision to publish, or preparation of the manuscript.

### Grant Disclosures

The following grant information was disclosed by the authors:

European Funds.

National Funds: PTDC/AGR-FOR/3807/2012 –FCOMP-01-0124-FEDER-027979.

FCT: UID/AMB/50017/2013.

FCT Investigator Programme: IF/00835/2013.

Post-doctoral grant to Liliana Santos: SFRH/BPD/90684/2012.

Ministerio de Ciencia e Innovación: BFU2010-17704.

Generalitat de Catalunya: 2009SGR809.

Dean for Research and the Departament de Ciències Mèdiques Bàsiques of the University of Lleida (Spain).

### Competing Interests

The authors declare there are no competing interests.

### Author Contributions

- Liliana Santos conceived and designed the experiments, performed the experiments, analyzed the data, contributed reagents/materials/analysis tools, wrote the paper, prepared figures and/or tables, reviewed drafts of the paper.
- Artur Alves conceived and designed the experiments, contributed reagents/materials/-analysis tools, wrote the paper, reviewed drafts of the paper.
- Rui Alves conceived and designed the experiments, analyzed the data, contributed reagents/materials/analysis tools, wrote the paper, prepared figures and/or tables, reviewed drafts of the paper.

### Data Availability

TreeBase (Study 20343): http://purl.org/phylo/treebase/phylows/study/TB2:S20343?format=html.

### Supplemental Information

Supplemental information for this article can be found online at http://dx.doi.org/10.7717/peerj.3120#supplemental-information.

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
