# Peer review of "Evaluating multi-locus phylogenies for species boundaries determination in the genus Diaporthe"

_PeerJ, doi:10.7717/peerj.3120_

## Round 0.1 · original submission · Minor Revisions

I recommend to you to submit the alignments and the trees to a database repository such as treebase. This will allow other authors working in same genus to use them as inputs for your sequences in an easier way.

·

Basic reporting

The information provided in this study is very relevant and help substantially to deciphering the Diaporthe’s black box through multi-loci phylogenies. The work was very well conducted and the manuscript is very well written. The manuscript is overall presented in an intelligible fashion. The authors demonstrated sufficient background in the topic. The structure of the article is adequate and the number of tables and figures are appropriate for the amount of data generated from this study. The literature provided by the authors covered all the taxonomical aspects of the genus Diaporthe, with many recent papers published in the subject.

The Fungal Barcoding Consortium (Schoch et al. 2012 PNAS 109: 6241–6246) ratified the ITS as the universal DNA barcode for the fungal kingdom using the same gene section proposed by White et al. (1990, book PCR protocols) more than 20 years earlier. Despite the strength and impact of rDNA ITS as the sanctioned universal fungal DNA barcode, its resolution of higher taxonomic level relationships is inferior to many protein-coding genes such as the TUB2. The most realistic solution would be to combine ITS with secondary or tertiary, ‘group-specific’ DNA barcode(s). This means combine the universal primer fidelity and high taxon coverage possible with ITS, with equally robust primers for secondary barcodes specific to the group or taxon of interest, enhancing precision of species identification.

Surprisingly, according to the authors the ITS locus is the least informative among the five loci tested in order to distinguish Diaporthe species. This finding is crucial for use in further taxonomic studies of this genus and will help scientists in the correct diagnosis of Diaporthe species. This will have important implications in the epidemiology and management of diseases caused by species of this genus.

Below some minor wording suggestions/corrections in the manuscript:

Line 42: consider changing ‘explosion’ by ‘increase’
Lines 44-45: it would be helpful for the reader to provide information about the recommendation of the use of Diaporthe over Phomopsis (Rossman et al. 2015 IMA Fungus 6:145-154). Many researchers still use the generic name Phomopsis.
Line 68: remove italics in ‘and’
Line 103: remove the first ‘was’
Line 260: rewrite ‘While Woese & Fox (1977) suggested…’
Line 293: ‘While Walker et al. (2012)…’
Line 313: ‘where we changed’
Line 343: ‘D. eres’ instead of ‘Diaporthe eres’
Line 346: resolution of this ‘species complex’
Line 351: rewrite ‘in the same clade could be due to the fact that …’

Experimental design

All the experiments led to solid conclusions. Experiments have been conducted rigorously, with appropriate replications.

Validity of the findings

The results are indeed interesting. The conclusions are drawn appropriately based on the data presented.

Reviewer 2 ·

Basic reporting

Overall writing is clear, but there are minor grammatical errors (subject verb, incorrect word usage)
Good background, references are thorough
Article structure is professional
Results are relevant to hypothesis

Experimental design

Novel analysis with published sequence data
Adressed an important question- which and how many loci are needed to speciate a certain group of fungi
Analyses were applied appropriately
Methods were thorough

Validity of the findings

replication of analyses, in terms of the hypothesized phylogenetic structure of the genus
Sequence data available from genbank and appropriate phylogenic analysis and model tests were completed
The conclusions in the Discussion are appropriate and not over stated
Speculation is minimal, and the paper rightly points out the lack of utility of ITS for fungal phylogenetic analysis when used alone.

Additional comments

There are grammatical errors that need to be fixed in an editorial fashion, and should not require additional review. These are relatively minor and do not confuse the reader.

I think trees and data should be submitted to treebase? I may have missed this.

---

## Round 0.2 · accepted · Accept

Thanks for taking into account all comments raised by both reviewers.